# The Clinical Features of In-Hospital Recurrence in Acute Ischaemic Stroke Patients over Time: A Real-World Observation at a Single Center

**DOI:** 10.3390/brainsci12020123

**Published:** 2022-01-18

**Authors:** Gaoqi Zhang, Qiong Yang, Huagang Zhang, Xiao Huang, Yu Fu, Dongsheng Fan

**Affiliations:** 1Department of Neurology, Peking University Third Hospital, Beijing 100191, China; gqzhang01@pku.edu.cn (G.Z.); yangqiongputh@126.com (Q.Y.); zhang-hg@bjmu.edu.cn (H.Z.); huangxiao@bjmu.edu.cn (X.H.); 2Key Laboratory for Neuroscience, National Health Commission/Ministry of Education, Peking University, Beijing 100191, China

**Keywords:** ischaemic stroke, antiplatelet therapy, risk factors

## Abstract

Acute ischaemic stroke (AIS) has a high risk of recurrence, particularly in the early stage. Our study aimed to assess the clinical characteristics and risk factors of in-hospital ischaemic recurrence in AIS patients in different periods. This study was a retrospective, single-center analysis. The patients were divided into two stages based on their admission time. The primary endpoint was recurrent stroke during hospitalization. In total, 978 patients in Stage 1 and 1047 patients in Stage 2 were included in this study. The in-hospital recurrence rate in Stage 1 was 5.9%, while that in Stage 2 was 4.0% (*p* = 0.046). A recurrence rate reduction mainly occurred in the minor stroke and large-artery atherosclerosis (LAA) stroke patients. Infection was an independent risk factor despite amelioration by antiplatelet therapy (*p* < 0.001). Diabetes patients also had a higher risk of in-hospital ischaemic recurrence among the minor stroke and large-artery atherosclerosis patients. A positive attitude towards antiplatelet therapy failed to completely halt recurrence of the disease. In conclusion, the rate of in-hospital ischaemic recurrence in AIS patients showed a decreasing trend over time, especially in the minor stroke and large-artery atherosclerosis stroke patients. Infection and diabetes were associated with a higher risk of stroke recurrence.

## 1. Introduction

Acute ischaemic stroke (AIS), the most common stroke subtype, is among the leading causes of disability and death worldwide [1]. Epidemiological studies have shown that AIS has a high risk of recurrence, particularly in the early stage. The cumulative rate of recurrent events reaches approximately 1.8% to 3.2% at 7 days [2,3,4], 3.1% to 4.2% at 30 days [5,6] and 3.7% to 5.7% at 90 days [7,8]. Among hospitalized patients, recurrence events may occur during their stay in the stroke unit, which prolongs the time of hospitalization and affects their prognosis. Previous studies have shown that the in-hospital recurrence rate of ischaemic stroke in AIS patients was 0.8% to 7.08% [9,10,11].

In 2013, the CHANCE (Clopidogrel in High-risk Patients with Acute Nondisabling Cerebrovascular Events) study [12] concluded that among patients with transient ischaemic attack (TIA) or minor stroke, the combination of clopidogrel and aspirin administered within 24 h after the onset of symptoms is superior to aspirin alone in reducing the risk of stroke in the first 90 days. In 2015, the “Guidelines for the Diagnosis and Treatment of Acute Ischaemic Stroke in China” regulated the standard antiplatelet therapy for AIS patients. The improvement in stroke treatments may contribute to a series of changes in the features of stroke recurrence. Thus, we aimed to assess the characteristics of in-hospital recurrent ischaemic stroke in different periods.

## 2. Materials and Methods

### 2.1. Study Population and Variables of Interest

Our study was a retrospective, single-center analysis based on patients admitted to the Neurology Department of Peking University Third Hospital between January 2014 and December 2017. The new version of the “Guidelines for the Diagnosis and Treatment of Acute Ischaemic Stroke in China” was published in 2015, as a result, the data were divided into 2 groups according to this timepoint. The patients in Stage 1 were admitted from January 2014 to December 2015, while the Stage 2 patients were admitted from January 2016 to December 2017.

Adult (≥18 years) patients were eligible for inclusion if they were admitted to the Neurology Department with acute ischaemic stroke (defined as ≤14 days). The diagnosis of the index stroke was confirmed by a trained neurologist based on the medical history, clinical presentation and neuroimaging results. Patients who were recently discharged from our or another hospital after a first TIA or stroke were not included in the study. Informed consent was obtained. The study was approved by the ethics committee of Peking University Third Hospital.

We documented the demographic information, medical history, vascular risk factors, clinical symptoms, stroke severity [assessed at admission using the National Institutes of Health Stroke Scale (NIHSS)] and aetiology of stroke according to the Trial of ORG 10172 in Acute Stroke Treatment (TOAST) criteria. We also recorded the auxiliary examinations (cerebrovascular imaging, carotid artery ultrasound, Holter and echocardiography), laboratory findings, systolic blood pressure level and treatment within 72 h after admission. All treatment options were guided by experienced neurologists. Glucose abnormalities were defined as fasting blood glucose ≤3.3 mmol/L or ≥10 mmol/L.

### 2.2. Outcome Measures

The primary endpoint was recurrent ischaemic stroke during hospitalization. Recurrent ischaemic stroke was defined as follows according to previous research [13]: (1) a new acute neurological event with symptoms lasting >24 h after the index stroke; (2) a new event occurring >24 h after the onset of the index stroke; and (3) an event not associated with other neurological or systemic causes, including but not limited to hemorrhagic transformation, oedema, seizure, heart failure, severe infection, etc. Clinical information for each patient was obtained from retrospective review using our electronic medical record system. All patients with a new acute neurological event underwent an imaging examination (noncontrast CT or MRI) to identify the cause. New neurological deficits caused by ischaemic lesions in different vascular territories were considered recurrent ischaemic events. An event was also considered a recurrent stroke if the new ischaemic lesion on the imaging was located in the same vascular region as the index stroke and was clearly associated with a new symptom. The outcome measurements were performed by two independent neurologists.

The other outcome events and complications included hemorrhagic transformation (HT), death, the modified Rankin scale (mRS) score at discharge and pulmonary or urinary infection. Hemorrhagic transformation was assessed by neuroimaging. Both symptomatic and asymptomatic HT were included [14]. The diagnosis of pulmonary or urinary infection was based on the medical records. The patients’ clinical presentations and laboratory or imaging examinations and the start of antibiotic therapy were carefully measured to confirm the diagnosis of infection.

### 2.3. Assessment of Antiplatelet Therapy

In our study, we assessed antiplatelet therapy during the hospital stay according to the NIHSS score and antiplatelet strategy. The definition of ‘minor stroke’ was based on the criterion in the CHANCE trial. We categorized antiplatelet treatment in noncardiac stroke patients into the following 3 groups: (1) the NEUTRAL group was defined as dual antiplatelet therapy (aspirin and clopidogrel) among minor stroke patients (NIHSS ≤ 3 points) or single antiplatelet therapy (aspirin or clopidogrel) among major stroke patients (NIHSS > 3 points); (2) the POSITIVE group was defined as DAPT (clopidogrel and aspirin) use in major stroke patients; and (3) the CONSERVED group was defined as single antiplatelet therapy (aspirin or clopidogrel) in minor stroke patients (NIHSS ≤ 3 points) without any contraindication in using DAPT. The patients who were not prescribed antiplatelet elements because of contraindications in drug use were assigned to the CONSERVED group. The contraindications of antiplatelet drugs included active bleeding events, thrombocytopenia and allergic reaction to aspirin or clopidogrel.

### 2.4. Statistical Analysis

The continuous variables are represented as the median and range, while the categorical variables are represented as numbers and percentages. A chi-square test was used to analyze the categorical variables, and a Mann–Whitney U test was used to evaluate the difference in the continuous variables. Logistic regression analysis models were used to evaluate the predictors associated with stroke recurrence and antiplatelet regimens. The results are represented using OR values and 95% confidence intervals. The variables that were statistically significant (*p* < 0.1) in the univariate analysis were included in the multivariate models. Statistical significance was defined as a two-sided *p*-value < 0.05. A Bonferroni correction was applied to correct for multiple comparisons. All analyses were performed using IBM SPSS Statistics (26.0) software.

## 3. Results

### 3.1. Patient Characteristics

In total, 2025 patients (978 patients in Stage 1 and 1047 patients in Stage 2) were included in this study. The baseline demographics and medical history of the patients at different timepoints are summarized in Table 1. Patients in different stages had no difference in age, sex, severity of index stroke (measured by the NIHSS score), history of ischaemic stroke, TIA, atrial fibrillation, hypertension or hyperlipidaemia. The patients admitted later were more likely to have diabetes. The proportion of large-artery atherosclerosis (LAA) stroke patients seemed to be slightly higher in Stage 2 (*p* = 0.013, no significant difference after the Bonferroni correction), while a significant decrease was observed in the undetermined-cause subtype (*p* < 0.001).

### 3.2. In-Hospital Ischaemic Recurrence in Different Stages

Among the patients, 100 patients (58 in Stage 1 and 42 in Stage 2) had a recurrent ischaemic event. A comparison of the demographic features, clinical characteristics and medical care between the recurrence and nonrecurrence groups during different periods is presented in Table 2.

The in-hospital recurrence rate in Stage 1 was 5.9%, while the rate in Stage 2 was 4.0% (*p* = 0.046). Figure 1A,B shows the time distribution of the recurrence events. Over 70% of the recurrent strokes occurred within the first 7 days during hospitalization (89.7% in Stage 1 and 71.4% in Stage 2). The time of the recurrence events was slightly later among the patients admitted later (Figure 2, *p* = 0.029). A reduction in the recurrence rate mainly occurred in the minor stroke (NIHSS ≤ 3 scores) and LAA patients (Figure 3A, *p* = 0.003; Figure 3B, *p* = 0.01).

Notably, the severity of the index stroke was greater in the patients with in-hospital recurrence in Stage 2, which manifested as a higher NIHSS score at admission (*p* < 0.001) and a greater possibility of an initial decrease in consciousness (*p* = 0.002). The comparison of medical care between the different groups is presented in Appendix A. More patients in Stage 2 received antiplatelet therapy, statins, and treatments for hypertension and diabetes. More patients underwent examinations, such as ultrasound cardiography (UCG) and Holter, over time, which revealed progress in cause screening and secondary prevention.

The in-hospital mortality among the patients with recurrent stroke was significantly higher than that among those without recurrence (*p* = 0.037 in Stage 1 and *p* = 0.024 in Stage 2). Furthermore, the length of hospital stay was significantly prolonged among the patients with in-hospital recurrence (*p* < 0.001 in both stages). The patients with recurrent stroke encountered a worse functional outcome at discharge (*p* < 0.001). The ratio of hemorrhagic events among the patients with recurrent stroke in the second period was higher than that in the first period (*p* = 0.004 vs. *p* = 0.578).

Univariable and multivariate regression analyses were performed to identify the risk factors associated with in-hospital recurrence (Appendix A and Table 3). The frequency of recurrent stroke was the lowest in the patients with small vessel disease in the combined analysis (*p* = 0.002). Although the aetiology of AA increased the risk of recurrent stroke in the first period, this result was not significantly reproduced in the later period (Table 1). The factors associated with diabetes and pulmonary or urinary infection were independent risk factors regardless of the admission period (*p* < 0.001), although the stability of the blood glucose level seemed to be managed better in Stage 2.

### 3.3. Clinical Characteristics of In-Hospital Recurrence in Different Subgroups

Given the reduction in the recurrence rate in the large-artery atherosclerosis (LAA) stroke and minor stroke (NIHSS ≤ 3 scores) patients, we aimed to analyze the features in these subgroups. In total, 946 LAA patients (429 in Stage 1 and 517 in Stage 2) and 1336 minor stroke patients (635 in Stage 1 and 701 in Stage 2) were included in the following analyses. Appendix A show the results of the univariate multinomial logistic regression analyses of the factors associated with in-hospital ischaemic recurrence in the abovementioned subgroups. Elements with statistical significance were included in the multivariate analyses, which are presented in Table 4. Infection was an independent risk factor for in-hospital recurrence in the LAA patients (*p* ≤ 0.001), while diabetes was another meaningful factor. Among the minor stroke patients, the blood pressure level seemed to be a consequence in Stage 1 (*p* = 0.019), while patients with recurrent ischaemic stroke were more likely to be prescribed anticoagulants in Stage 2 (*p* = 0.001). Surprisingly, the influence of infection was not significant among the minor stroke patients during Stage 1.

### 3.4. Factors Associated with Antiplatelet Therapy in Noncardiac Stroke Patients

A reduction in the in-hospital recurrence rate mainly occurred in the large-artery atherosclerotic stroke and minor stroke patients, who constitute the target population of antiplatelet therapy. As a result, we analyzed the treatment trends of antiplatelet therapy in our center (Table 5). The patients who underwent recurrent stroke seemed to be treated more aggressively in Stage 2, although the result was not significant after the correction. A univariate multinomial logistic regression analysis of the factors affecting antiplatelet regimens was also carried out (Table 6). Elderly people (≥75 years) and small vessel stroke patients were likely to be treated conservatively (*p* < 0.001), while the patients with diabetes (*p* = 0.002) and patients who presented with paresis or ataxia (*p* < 0.001) were less likely to be undertreated. Patients prescribed statins or hypoglycemic agents might be less likely to be undertreated (*p* < 0.001). Large-artery atherosclerosis stroke and symptoms associated with large vascular disease (aphasia or disturbance of consciousness) led to a trend of positive treatment. A multivariate analysis further showed that age and clinical manifestations are relatively more important than others (Appendix A).

## 4. Discussion

This research is a retrospective, real-world analysis focusing on recurrent stroke in AIS patients in our in-hospital cohort. Our study found that the in-hospital recurrence rate was reduced over time, especially in the minor stroke patients and large-artery atherosclerosis stroke patients. Infection was an obvious risk factor for in-hospital stroke recurrence. Diabetes may also be an influencing factor, especially in large-artery atherosclerosis stroke groups. A tendency of positive antiplatelet therapy was also presented over time.

Our results show that a decrease in in-hospital recurrence mainly occurred in patients with the LAA subtype. There is a consensus that LAA patients have a higher risk of early recurrence similar to the patients in our center during the early stage. However, we reached a different conclusion regarding patients admitted during the later stage (Table 1, *p* < 0.001 in Stage 1 vs. *p* = 0.097 in Stage 2), although the absolute value of the recurrence rate was still higher in the patients with the LAA subtype. The subgroup analysis also revealed a reduction in the recurrence rate in the minor stroke patients. Increasing evidence suggests that DAPT with clopidogrel and aspirin provides greater protection against recurrent stroke than monotherapy, especially in minor stroke patients and patients with arterial stenosis. The CHANCE trial [12] published in 2013 demonstrated the benefit of DAPT within 24 h of symptom onset in Chinese minor stroke patients despite the increased risk of bleeding. Subsequently, the POINT trial in 2018 [15], which included multiracial patients across various centers, reported reductions in ischaemic stroke occurrence in the DAPT cohort. The THALES trial in 2020 extended the DAPT regimen as the combination of ticagrelor and aspirin could reduce the risk of the composite of stroke or death within 30 days in mild-to-moderate AIS patients [16]. Nevertheless, in our study, the reduction in in-hospital recurrence was not related to DAPT, although the patients in those subgroups should have theoretically benefited from DAPT. Further analysis of the tendency of antiplatelet regimens showed that the patients with recurrence received more positive therapy. The patients with clinical features associated with atherosclerosis also received active antiplatelet treatment. Due to the limitations of retrospective studies, this result can only reveal a relationship in which patients with a higher risk of recurrent ischaemic stroke may also be treated more aggressively.

Our research also shows progress in comprehensive stroke management. The new version of Chinese stroke guideline pointed out the importance of treatment in acute phase. The guideline standardized the extended time window, indications and contraindications for intravenous thrombolysis. Apart from the early use of dual antiplatelet therapy, other treatments including anticoagulant therapy, blood pressure management, blood glucose management and the use of neuroprotective agents were also updated. In our center, the patients received more aggressive treatments and risk factor screening examinations in the later period. Although statins do not appear to be effective in preventing all types of strokes or reducing all-cause mortality, they might reduce the risk of recurrent ischaemic events with previous attacks [17]. In-hospital statin initiation was linked to better early stroke outcomes [18]. The early initiation of existing treatments, including antiplatelet therapy, statins and blood pressure control, after minor stroke could result in an 80% reduction in the risk of early recurrence [19]. Some drug trials have revealed a beneficial effect of intensive glycemic control on risk of stroke although further high-quality studies are required to confirm that [20]. Therefore, the clinical practice in our center reflected the effects of improved secondary prevention on the decrease in in-hospital recurrence. However, the number of patients with reperfusion therapy is limited. Improvement in reperfusion therapy is also not obvious in our center. Although the current study did not show any association between early reperfusion therapy and in-hospital recurrence in atherosclerotic stroke or minor stroke patients, reperfusion is still the primary goal of stroke treatment. Therefore, promotion of reperfusion therapy should be prioritized for improvement in future development efforts.

Despite the improvement in the awareness of secondary prevention, many patients had in-hospital recurrences. An alternative reason is the presence of antiplatelet nonresponsiveness. For example, polymorphisms in the CYP2C19 gene are strongly associated with the therapeutic effect of clopidogrel, and loss-of-function variants in the CYP2C19 gene are widespread in East Asian populations [21]. Unfortunately, due to data deficiency in our center, these factors were not included in this analysis. Infection is another noteworthy topic. Yu F et al. [9], Erdur et al. [10] and Xu et al. [22] analyzed the risk factors of in-hospital recurrence and noted that urinary or respiratory infections were associated with a higher risk of stroke recurrence. Infection may trigger the recurrence of stroke via infection-related platelet activation and aggregation, inflammation-induced thrombosis, impaired endothelial function, etc. [23]. The glycoprotein (GP) Ib and GPVI-mediated pathways and the activation of coagulation factor XII, rather than GPIIb–IIIa-mediated aggregation, are the main checkpoints in inflammation-induced platelet activation [24]. The current antiplatelet drugs have limited influence on the pathways mentioned above. Therefore, the common treatments may not be effective enough to prevent neothrombosis. However, the preventive use of antibiotic therapy in AIS patients cannot improve the clinical outcomes despite a reduction in urinary tract infections [25]. Moreover, neuroinflammation has been recognized as a critical element in the onset and progression of ischaemic stroke. The role of platelet-immunocyte interactions during inflammation also contributes to atherosclerosis [26]. Further studies should focus on biomarkers and interventions in thrombotic and neuroinflammatory pathways. The burden of diabetes is another important factor related to early recurrent stroke. A post hoc analysis of the CHANCE trial showed that diabetes was associated with an increased risk of recurrent stroke after a minor stroke or TIA after 3 months of follow-up. There was no difference in the effect of antiplatelet treatment in reducing these events in patients with or without diabetes [27].

Our stratification of antiplatelet regimens was mainly based on the criterion used in the CHANCE trial, which was a coarse analysis. However, a trend of radical antiplatelet therapy was shown during the period of the guideline update. Kim et al. [28] reported that a large atherosclerotic burden might affect the selection of a combination of DAPT. Similarly, the large-artery atherosclerosis patients in our study were likely to be treated positively. Patients who present with aphasia or coma may have intracranial or external vascular stenosis; thus, these patients may be treated aggressively even before a vascular examination. A previous study reported that the risk reduction in the subgroup with diabetes by antiplatelet treatment was only 7% compared with an average reduction of 22% in all patients [29]. The patients with unstable blood glucose in our study were less likely to be undertreated, probably to prevent any form of deterioration, including recurrence. However, a potential risk of positive antiplatelet therapy exists. A post hoc analysis of the CHANCE trial revealed that compared with monotherapy, the benefit of clopidogrel-aspirin treatment was offset by the potential risk of hemorrhage after a short course of treatment, which was approximately 10 days [30]. The rate of bleeding events increased during the second period in our center, although these patients already had a higher risk of hemorrhagic transformation considering the stroke subtype and severity of the index stroke (Appendix A).

Another aspect worthy of attention is that patients with cardioembolic stroke have a high risk of in-hospital recurrence during both periods. The risk of early recurrence in the first two weeks after cardioembolic ischaemic stroke is between approximately 0·5% and 1·3% per day [31]. It seems that anticoagulant therapy is more widely used in our center, but there was no statistically significant decrease in recurrence. A previous study revealed that ischaemic stroke patients with atrial fibrillation shared similar risk factors with thrombotic stroke patients [32]. Meanwhile, the risk factors for atherosclerosis, including diabetes and hypertension, are responsible for the development of atrial fibrillation. Infection is also related to atrial fibrillation in stroke patients [33]. Therefore, potentially undiagnosed atrial fibrillation may also contribute to in-hospital recurrence. This finding highlights that long-term electrocardiographic (ECG) monitoring needs to be more widely adopted.

This study has some limitations. First, our study adopted a retrospective, single-center design and lacked out-of-hospital follow-up information. Our findings need to be further confirmed in a study with a larger sample size and prospective research. Second, the assessment of antiplatelet therapy adopted a crude approach. Vascular evaluation results are also important in choosing an antiplatelet regimen, and these data were not included during the grouping process. Third, we could only use the information in electronic medical records; as a result, some important information, such as the mechanism of recurrent stroke, was not included. The evaluations of other risk factors, treatments, comorbidities and complications are also insufficient. In Stage 2, the prescription of anticoagulants seemed to be related to ischaemic recurrence. However, accurate records of anticoagulation treatment after antiplatelet therapy are not available for all cardioembolic stroke cases; thus, we did not analyze the relationship between treatments and in-hospital recurrence in cardioembolic patients. Compared with Stage 1, more patients in Stage 2 received antihypertensive or hypoglycemic treatment. Further analysis needs to be broadened to include a more detailed therapy regimen and the effect on the outcome event. Fourth, the lack of detailed imaging data and vascular assessment results is another important limitation. Infarcts in different periods, multiple acute infarcts, lesions with different circulations, and isolated cortical lesions all indicate a higher risk of early recurrence [8]. Minor stroke patients with acute large vessel occlusion are at risk of early recurrence, which may be caused by hypoperfusion [34]. CTP changes can be used to predict subsequent ischaemic tissue injury on DWI in TIA/minor stroke patients and have been shown to be of value in predicting recurrence [35]. The lack of relevant evaluation may lead to under-identification of risk populations. This population may have been undertreated in our study. The lack of advanced imaging may also deny a segment of patients access to a potentially beneficial reperfusion treatment. Therefore, the imaging features of the index stroke and evaluation of vascular stenosis should strongly be considered in subsequent studies.

## 5. Conclusions

This study showed a decreasing trend in in-hospital recurrence in Chinese ischaemic stroke patients despite the relatively high recurrence rate. The reduction mainly occurred in large-artery atherosclerosis stroke and minor stroke patients. The subgroup analyses revealed that infection and diabetes were associated with recurrence regardless of the reduction in the recurrence rate. Positive antiplatelet therapy failed to prevent all occurrences of in-hospital recurrence.

## Figures and Tables

**Figure 1 brainsci-12-00123-f001:**
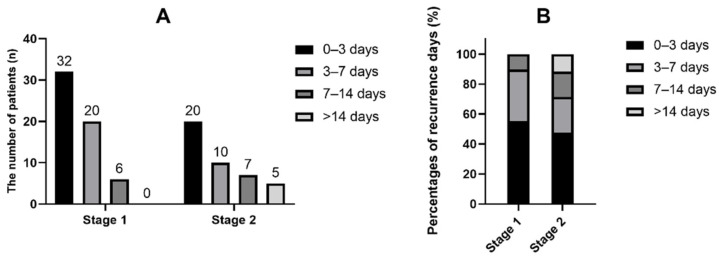
The time profile (**A**) and distribution (**B**) of recurrence events in the two periods.

**Figure 2 brainsci-12-00123-f002:**
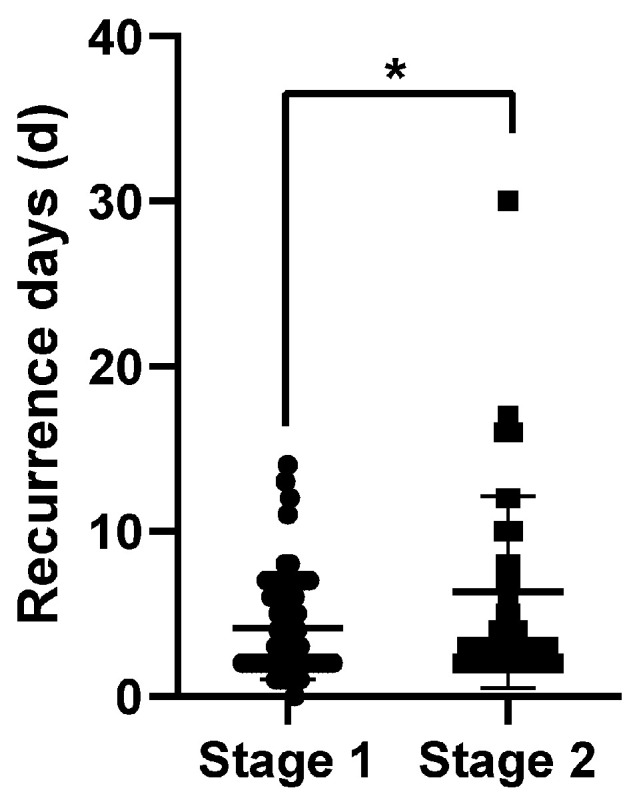
Comparison of recurrence days between different periods. *—significant difference.

**Figure 3 brainsci-12-00123-f003:**
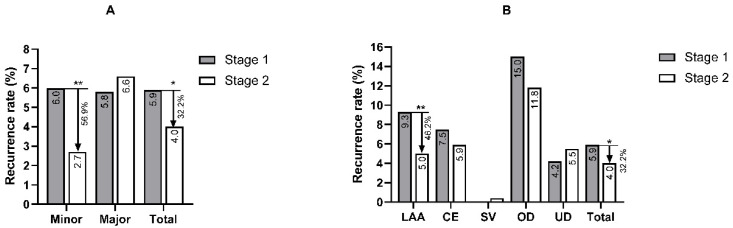
The recurrence rate in patients with different stroke severities (**A**) and TOAST subtypes (**B**). TOAST, Trial of Org 10172 in Acute Stroke Treatment; Minor, minor stroke (NIHSS score ≤ 3 points); Major, major stroke (NIHSS > 3 points); AA, artery atherosclerosis; CE, cardio-aortic embolism; SV, small vessel disease; OD, other determined causes; UD, undetermined causes. The levels of significance (*p*) in each subgroup were equal to 0.025 (0.05/2) in Figure 2A and 0.01 (0.05/5) in Figure 2B. *—significant difference (*p* < 0.05), **—significant difference (*p* < 0.01).

**Table 1 brainsci-12-00123-t001:** Baseline demographics and medical history of the patients in different stages.

	Stage 1 (n = 978)	Stage 2 (n = 1047)	*p*-Value
Age (years)	64 (25–98)	63 (20–93)	0.954
Gender (Male, %)	722 (73.8)	786 (75.1)	0.520
**Medical history**			
Previous IS (%)	232 (23.7)	212 (20.2)	0.059
Previous TIA (%)	64 (6.5)	78 (7.4)	0.425
Atrial fibrillation (%)	89 (9.1)	96 (9.2)	0.957
Hypertension (%)	668 (68.3)	740 (70.7)	0.246
Diabetes (%)	336 (34.4)	405 (38.7)	0.043
Hyperlipidaemia (%)	210 (21.5)	229 (21.9)	0.819
**TOAST**			
AA (%)	429 (43.9)	517 (49.4)	0.013 ^#^
CE (%)	80 (8.2)	85 (8.1)	0.960
SV (%)	233 (23.8)	282 (26.9)	0.108
OD (%)	20 (2.0)	17 (1.6)	0.479
UD (%)	213 (21.8)	146 (13.9)	<0.001 *
**NIHSS score**	2 (0–27)	2 (0–36)	0.140

*—significant difference after the Bonferroni correction (*p* < 0.01). ^#^—significant difference before the correction. IS, ischaemic stroke; TIA, transient ischaemic attack; TOAST, Trial of Org 10172 in Acute Stroke Treatment; AA, artery atherosclerosis; CE, cardio-aortic embolism; SV, small vessel disease; OD, other determined causes; UD, undetermined causes; NIHSS, National Institutes of Health Stroke Scale.

**Table 2 brainsci-12-00123-t002:** Comparison of the demographic features, clinical characteristics and medical care between the recurrence and nonrecurrence groups in different stages.

	Stage 1 (n = 978)			Stage 2 (n = 1047)		
	Nonrecurrence (n = 920)	Recurrence (n = 58)	*p*-Value	Nonrecurrence (n = 1005)	Recurrence (n = 42)	*p*-Value
Age (years)	63 (25–98)	68 (29–85)	0.106	63 (20–93)	63.5 (27–92)	0.418
Gender (male, %)	676 (73.5)	46 (79.3)	0.306	754 (75.0)	32 (78.2)	0.864
**Medical history**						
Previous IS (%)	221 (24.0)	11 (19.0)	0.380	201 (20.0)	11 (26.2)	0.328
Previous TIA (%)	56 (6.1)	8 (13.8)	0.047 *	74 (7.4)	4 (9.5)	0.547
Atrial fibrillation (%)	82 (8.9)	7 (12.1)	0.418	88 (8.8)	8 (19.0)	0.048 *
Hypertension (%)	627 (68.2)	41 (70.7)	0.687	709 (70.5)	31 (73.8)	0.649
Diabetes (%)	309 (33.6)	27 (46.6)	0.044 *	381 (37.9)	24 (57.1)	0.012 *
Hyperlipidaemia (%)	192 (20.9)	18 (31.0)	0.067	219 (21.8)	10 (23.8)	0.759
**Presentation**						
Paresis/ataxia (%)	643 (69.9)	43 (74.1)	0.556	875 (87.1)	38 (90.5)	0.657
Aphasia (%)	332 (36.1)	12 (20.7)	0.017 *	137 (13.6)	10 (23.8)	0.063
Coma (%)	28 (3.0)	0 (0.0)	0.404	52 (5.2)	8 (19.0)	0.002 *
**NIHSS score**	2 (0–27)	3 (0–12)	0.767	2 (0–36)	4.5 (0–28)	<0.001 *
**TOAST**						
AA (%)	389 (42.3)	40 (60.9)	<0.001 *	491 (48.9)	26 (61.9)	0.097
CE (%)	74 (8.0)	6 (10.3)	0.464	80 (8.0)	5 (11.9)	0.359
SV (%)	233 (25.3)	0 (0.0)	<0.001 *	281 (28.0)	1 (2.4)	<0.001 *
OD (%)	17 (1.8)	3 (5.2)	0.110	15 (1.5)	2 (4.8)	0.146
UD (%)	204 (22.2)	9 (15.5)	0.324	138 (13.7)	8 (19.0)	0.330
**Indicators**						
HCY (µmol/L)	15.21 (0.2–130.9)	14.97 (8.61–121.6)	0.877	12.21 (4.97–109.52)	11.34 (5.68–36.54)	0.111
hsCRP (mg/L)	2.05 (0.01–209.88)	2.01 (0.46–82.23)	0.370	1.40 (0.01–120.86)	3.73 (0.12–78.37)	<0.001 *
CHL (mmol/L)	4.19 (1.41–11.25)	3.71 (2.53–7.36)	0.256	3.95 (1.99–7.68)	3.94 (2.25–8.94)	0.152
TG (mmol/L)	1.38 (0.39–13.65)	1.38 (0.61–4.16)	0.905	1.37 (0.35–14.52)	1.32 (0.47–15.61)	0.382
LDL (mmol/L)	2.48 (0.43–5.50)	2.19 (1.07–5.96)	0.286	2.41 (0.73–5.41)	2.33 (1.14–4.93)	0.152
HDL (mmol/L)	0.96 (0.41–3.74)	0.97 (0.62–1.73)	0.703	0.95 (0.47–3.22)	0.89 (0.58–1.52)	0.304
FBG (mmol/L)	5.10 (2.30–21.23)	5.45 (4.0–12.00)	0.019 *	5.6 (3.2–20.2)	6.1 (3.2–15.6)	0.066
HbA1C (%)	5.8 (4.2–15.6)	6.05 (4.7–13.5)	0.012 *	6.1 (4.6–14.2)	6.5 (5.0–12.0)	0.136
Glucose abnormality (%)	67 (7.3)	14 (24.1)	<0.001 *	101 (10.2)	7 (16.7)	0.192
Average BP (mmHg)	138.0 (94.0–190.8)	143.0 (113.0–170.8)	0.011 *	142.0 (99.3–201.2)	140.7 (109.6–177.5)	0.764
BPSD	9.57 (0–57.44)	11.2 (1.9–23.2)	0.019 *	10.5 (0.96–59.4)	11.5 (4.03–27.1)	0.162
**Treatment**						
IVT (%)	83 (9.0)	4 (6.9)	0.580	50 (5.0)	2 (4.8)	0.956
AntiPlt (%)	884 (96.1)	55 (94.8)	0.499	961 (95.6)	40 (95.2)	0.707
DAPT (%)	360 (39.1)	28 (48.3)	0.167	684 (68.1)	27 (64.3)	0.608
Anticoagulant (%)	31 (3.4)	4 (6.9)	0.148	83 (8.3)	9 (21.4)	0.008 *
AntiHTN (%)	447 (48.6)	27 (46.6)	0.764	641 (63.8)	25 (59.5)	0.574
AntiDM (%)	255 (27.7)	21 (36.2)	0.164	353 (35.1)	20 (47.6)	0.098
Statin (%)	835 (90.8)	55 (94.8)	0.474	988 (98.3)	41 (97.6)	0.524
EVT in 24 h (%)	5 (0.5)	2 (3.4)	0.060	12 (1.2)	0 (0.0)	1.000
**Examination**						
Holter (%)	75 (8.2)	4 (6.9)	1.000	517 (51.4)	26 (61.9)	0.408
UCG (%)	855 (92.9)	54 (93.1)	1.000	954 (94.9)	41 (97.6)	0.521
CTA/MRA/DSA (%)	818 (88.9)	50 (86.2)	0.527	957 (95.2)	41 (97.6)	0.717
CVUS (%)	839 (91.2)	56 (96.6)	0.222	936 (93.1)	40 (95.2)	0.859
**Outcome**						
LOS (days)	14 (1–94)	18 (3–40)	<0.001 *	14 (2–90)	20.5 (4–89)	<0.001 *
Mortality (%)	10 (1.1)	3 (5.2)	0.037 *	13 (1.3)	3 (7.1)	0.024 *
Pulmonary or urinary infection (%)	68 (7.4)	10 (17.2)	0.020 *	77 (7.7)	19 (45.2)	<0.001 *
Haemorrhage (%)	13 (1.4)	1 (1.7)	0.578	23 (2.3)	5 (11.9)	0.004 *
mRS (0–2, %)	823 (89.5)	41 (74.5)	<0.001 *	877 (88.3)	14 (36.8)	<0.001 *

*—significant difference after the Bonferroni correction (*p* < 0.01). IS, ischaemic stroke; TIA, transient ischaemic attack; NIHSS, National Institutes of Health Stroke Scale; TOAST, Trial of Org 10172 in Acute Stroke Treatment; AA, artery atherosclerosis; CE, cardio-aortic embolism; SV, small vessel disease; OD, other determined causes; UD, undetermined causes; IVT, intravenous thrombolysis; AntiPlt, antiplatelet; DAPT, dual antiplatelet therapy; AntiHTN, anti-hypertension agents; AntiDM, anti-diabetes agents; EVT, endovascular treatment; UCG, ultrasonic cardiogram; CTA, computed tomography angiography; MRA, magnetic resonance angiography; DSA, digital subtraction angiography; CVUS, carotid vessel ultrasound. HCY, homocysteine; hsCRP, hypersensitive C-reactive protein; CHL, cholesterol; TG, triglyceride; LDL, low-density lipoprotein; HDL, high-density lipoprotein; FBG, fasting blood glucose; HbA1C, glycosylated haemoglobin; BP, blood pressure; BPSD, standard deviation of blood pressure; LOS, length of stay; mRS: modified Rankin score.

**Table 3 brainsci-12-00123-t003:** Multivariate analysis of the factors associated with recurrent stroke in different stages.

	Multivariate Analysis			
	OR	95% CI		*p*-Value
Stage 1				
Aphasia	0.394	0.200	0.775	0.007
Infection	2.566	1.155	5.702	0.021
Glucose abnormality	3.055	1.542	6.055	0.001
TOAST				
UD	1.000			
AA	2.254	1.062	4.786	0.034
Stage 2				
Diabetes	1.998	1.032	3.868	0.040
Pulmonary or urinary infection	9.856	4.918	19.750	<0.001
TOAST				
UD	1.000			
SV	0.059	0.007	0.492	0.009
Combined				
Pulmonary or urinary infection	4.658	2.843	7.631	<0.001
Glucose abnormality	2.572	1.514	4.371	<0.001
TOAST				
UD	1.000			
SV	0.043	0.006	0.327	0.002

TOAST, Trial of Org 10172 in Acute Stroke Treatment; AA, artery atherosclerosis; SV, small vessel disease; OR, odds ratio; CI, confidence interval.

**Table 4 brainsci-12-00123-t004:** Multivariate analysis of the factors associated with recurrent stroke in different subgroups.

Large-Artery Atherosclerosis					Minor Stroke				
	OR	95% CI		*p*-Value		OR	95% CI		*p*-Value
Stage 1					Stage 1				
Pulmonary or urinary infection	4.967	2.622	9.410	<0.001	Aphasia	0.351	0.130	0.947	0.039
Glucose abnormality	3.300	1.789	6.085	<0.001	Glucose abnormality	2.505	1.047	5.989	0.039
Stage 2					Average BP	1.029	1.005	1.054	0.019
Diabetes	3.050	1.265	7.352	0.013	Stage 2				
NIHSS score	1.114	1.016	1.222	0.022	Pulmonary or urinary infection	12.078	3.378	43.191	<0.001
Pulmonary or urinary infection	5.605	2.044	15.367	0.001	Anticoagulants	6.419	2.237	18.419	0.001
Combined					Combined				
Pulmonary or urinary infection	4.967	2.622	9.410	<0.001	Previous TIA	2.622	1.254	5.483	0.010
Glucose abnormality	3.300	1.789	6.085	<0.001	Pulmonary or urinary infection	4.161	1.791	9.667	0.001
					Anticoagulant	4.252	1.895	9.541	<0.001
					Glucose abnormality	3.002	1.495	6.026	0.002
					AverageBP	1.020	1.001	1.039	0.041

TIA, transient ischaemic attack; NIHSS, National Institutes of Health Stroke Scale; BP, blood pressure; OR, odds ratio; CI, confidence interval.

**Table 5 brainsci-12-00123-t005:** The relationship between antiplatelet therapy and in-hospital recurrence in noncardiac stroke patients in different stages.

	Combined (n = 1860)				Stage 1 (n = 898)				Stage 2 (n = 862)			
Treatment Trends	OR	95% CI		*p*-Value	OR	95% CI		*p*-Value	OR	95% CI		*p*-Value
Conserved	1.000				1.000				1.000			
Neutral	1.211	0.714	2.056	0.477	1.499	0.803	2.798	0.204	1.412	0.472	4.227	0.538
Positive	1.871	1.019	3.437	0.043 ^#^	1.369	0.576	3.252	0.477	3.511	1.143	10.780	0.028 ^#^

^#^—significant difference before the correction. No significant difference after the Bonferroni correction (*p* < 0.025); OR, odds ratio; CI, confidence interval.

**Table 6 brainsci-12-00123-t006:** Univariate analysis of the determinants associated with the antiplatelet regimen choice in noncardiac stroke patients.

	Conserved				Positive			
Factor	OR	95% CI		*p*-Value	OR	95% CI		*p*-Value
Age (≥75 years)	1.613	1.264	2.058	<0.001 *	0.886	0.642	1.223	0.462
Previous IS	1.013	0.783	1.310	0.922	1.357	1.016	1.813	0.038 ^#^
Previous TIA	1.013	0.683	1.502	0.948	0.742	0.441	1.249	0.262
Hypertension	0.865	0.692	1.083	0.206	1.137	0.862	1.500	0.365
Diabetes	0.701	0.562	0.874	0.002 *	1.106	0.859	1.425	0.433
History of hyperlipaemia	0.940	0.729	1.211	0.940	0.995	0.738	1.342	0.995
Paresis/ataxia	0.547	0.431	0.696	<0.001 *	3.151	2.021	4.914	<0.001 *
Aphasia	0.826	0.637	1.070	0.148	1.466	1.109	1.938	0.007 *
Coma	0.301	0.116	0.782	0.014 *	2.668	1.539	4.624	<0.001 *
AA	0.632	0.512	0.781	<0.001 *	1.348	1.047	1.737	0.021 *
SV	1.452	1.158	1.820	0.001 *	0.686	0.506	0.930	0.015 *
IVT	1.007	0.660	1.537	0.973	0.975	0.586	1.623	0.922
AntiHTN	0.874	0.709	1.079	0.211	1.110	0.862	1.430	0.419
AntiDM	0.646	0.512	0.816	<0.001 *	1.215	0.940	1.571	0.137
Statin	0.180	0.107	0.303	<0.001 *	1.364	0.508	3.663	0.538

*—significant difference after the Bonferroni correction (*p* < 0.025). ^#^—significant difference before the correction. IS, ischaemic stroke; TIA, transient ischaemic attack; AA, artery atherosclerosis; SV, small vessel disease IVT, intravenous thrombolysis; AntiHTN, anti-hypertension agents; AntiDM, anti-diabetes agents; OR, odds ratio; CI, confidence interval.

## Data Availability

The data presented in this study are available upon request from the corresponding author. The data are not publicly available due to data management regulations in our hospital.

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
