# Peer review of "The Clinical Features of In-Hospital Recurrence in Acute Ischaemic Stroke Patients over Time: A Real-World Observation at a Single Center"

_brainsci, 2022, doi:10.3390/brainsci12020123_

Round 1
Reviewer 1 Report
The manuscript presents intersting findings considering factors associated of in-hospital recurrence in patients with acute ischemic stroke. However, it is not flawless. In particular, the following issues need to be addressed:
- the authors should do careful spelling check as the manuscript is full of misspelings and typos, like oedemaedema, aphsia, etc.;
- the authors should desscibe in more details the differences between Stage 1 and Stage 2, apart from the date of patients' enrollment;
- the date and protocol number of ethics committee should be mentioned;
- what was the definition of "infection"? any type, including conjunctivitis?
Author Response
Dear reviewer, thank you for reviewing our manuscript and for the constructive comments. We have revised the manuscript (Manuscript ID: brainsci-1515981). Here are the point-by-point responses for the reviewers’ comments:
Comment 1: the authors should do careful spelling check as the manuscript is full of misspelings and typos, like oedemaedema, aphsia, etc.;
Response to comment 1: Thank you for your kind patients. We have checked the spellings and the revised manuscript has been polished by a professional academic polishing company.
Comment 2: the authors should describe in more details the differences between Stage 1 and Stage 2, apart from the date of patients' enrollment;
Response to comment 2: Thank you for your valuable comment. We aimed to assess the clinical and treatment characteristics of in-hospital recurrent ischaemic stroke in different periods. The update of the clinical guidelines and standardized treatment may improve the in-hospital outcomes in AIS patients. As a result, one of the purposes of the research is to analyse the treatment in different periods, especially before and after guideline update. As the updated version of the “Guidelines for the Diagnosis and Treatment of Acute Ischaemic Stroke in China” was published in the second half of the year 2015, the grouping of the patients was determined by that. Baseline information and treatment comparison are shown in the form of a table in the revised manuscript (Table 1 and Table S1).
Table 1. Baseline demographics and medical history of the patients in different stages.
|
|
Stage 1 (n=978) |
Stage 2 (n=1047) |
P value |
|
Age (years) |
64 (25-98) |
63 (20-93) |
0.954 |
|
Gender (Male, %) |
722 (73.8) |
786 (75.1) |
0.520 |
|
Medical history |
|
|
|
|
Previous IS (%) |
232 (23.7) |
212 (20.2) |
0.059 |
|
Previous TIA (%) |
64 (6.5) |
78 (7.4) |
0.425 |
|
Atrial fibrillation (%) |
89 (9.1) |
96 (9.2) |
0.957 |
|
Hypertension (%) |
668 (68.3) |
740 (70.7) |
0.246 |
|
Diabetes (%) |
336 (34.4) |
405 (38.7) |
0.043 |
|
Hyperlipidaemia (%) |
210 (21.5) |
229 (21.9) |
0.819 |
|
TOAST |
|
|
|
|
AA (%) |
429 (43.9) |
517 (49.4) |
0.013# |
|
CE (%) |
80 (8.2) |
85 (8.1) |
0.960 |
|
SV (%) |
233 (23.8) |
282 (26.9) |
0.108 |
|
OD (%) |
20 (2.0) |
17 (1.6) |
0.479 |
|
UD (%) |
213 (21.8) |
146 (13.9) |
<0.001* |
|
NIHSS score |
2 (0-27) |
2 (0-36) |
0.140 |
* - significant difference after Bonferroni correction (p<0.01). #-significant difference before correction. IS, ischaemic stroke; TIA, transient ischaemic attack; TOAST, the Trial of Org 10172 in Acute Stroke Treatment; AA, artery atherosclerosis; CE, cardio-aortic embolism; SV, small vessel disease; OD, other determined causes; UD, undetermined causes; NIHSS, the National Institutes of Health Stroke Scale.
Table S1. The comparison of treatment and auxiliary examination between different stages
|
|
Stage 1 (n=978) |
Stage 2 (n=1047) |
P value |
|
Treatment |
|
|
|
|
IVT (%) |
87 (8.9) |
52 (5.0) |
0.001* |
|
EVT in 24h (%) |
7 (0.7) |
12 (1.1) |
0.315 |
|
AntiPlt (%) |
939 (96.0) |
1001 (95.6) |
<0.001 |
|
None (%) |
39 (4.0) |
46 (4.4) |
0.649 |
|
Mono (%) |
551 (56.3) |
290 (27.7) |
<0.001* |
|
Dual (%) |
388 (39.7) |
711 (67.9) |
<0.001* |
|
Anticoagulant (%) |
35 (3.6) |
92 (8.8) |
<0.001* |
|
AntiHTN (%) |
474 (48.5) |
666 (63.6) |
<0.001* |
|
AntiDM (%) |
276 (28.2) |
373 (35.6) |
<0.001* |
|
Statin (%) |
890 (91.0) |
1028 (98.2) |
<0.001* |
|
Examination |
|
|
|
|
Holter (%) |
79 (8.1) |
543 (51.9) |
<0.001* |
|
UCG (%) |
909 (92.9) |
995 (95.0) |
0.048* |
|
CT (%) |
815 (83.3) |
597 (57.0) |
<0.001* |
|
MRI (%) |
940 (96.1) |
981 (93.7) |
0.014* |
|
CTA/MRA/DSA (%) |
868 (88.8) |
998 (95.3) |
<0.001* |
|
CVUS (%) |
895 (91.5) |
976 (93.2) |
0.148 |
* - significant difference (Bonferroni-adjusted P<0.0167 for AntiPlt). IVT, intravenous thrombolysis; EVT, endovascular treatment; AntiPlt, antiplatelet; AntiHTN, anti-hypertension agents; AntiDM, anti-diabetes agents; UCG, ultrasonic cardiogram; CTA, computed tomography angiography; MRA, magnetic resonance angiography; DSA, digital subtraction angiography; CVUS, carotid vessel ultrasound.
Comment 3: the date and protocol number of ethics committee should be mentioned;
Response to comment 3: Thank you for your grateful indication. Our research was approved by the Medical Science Research Ethics Committee of Peking University Third Hospital (IRB00006761-M2019439). We have also added the above relevant information in the revised manuscript (Institutional Review Board Statement part).
Comment 4: what was the definition of "infection"? any type, including conjunctivitis?
Response to comment 4: Thank you for your kind reminder. The “infection” in our study referred to pulmonary or urinary infection. The diagnosis of pulmonary or urinary infection was based on medical records. A clear diagnosis of pulmonary or urinary infection and beginning of antimicrobial therapy were used as references. Related narrative was presented in the Materials and Methods part. The descriptions are as follows: “The diagnosis of pulmonary or urinary infection was based on medical records. Patients’ clinical presentations, laboratory or imaging examinations and the start of antibiotic therapy were carefully measured to confirm the diagnosis of infection”. We also added detailed description in the main text to avoid unnecessary misunderstanding.
Reviewer 2 Report
The article is well written and brings new knowledge to the issue of ischemic stroke related to the thrombotic mechanism.
This study is carried out on a large group of patients, which increases its scientific value
Section: Materials and Methods, Results do not require correction.
Nevertheless, in the introduction or discussion section:
a) it is worth mentioning embolic strokes, which in most cases result from the same risk factors as thrombotic strokes.
https://doi.org/10.5114/aoms.2019.84212
b) it is worth referring to the manuscript which emphasizes the importance of statins in relation to patients with ischemic stroke and atrial fibrillation.
Author Response
Dear Reviewer: Thank you for your comments and providing good suggestions for my manuscript. We have revised my manuscript (Manuscript ID: brainsci-1515981) according to your and reviewer's suggestion. My explanation to the comments point- by- point is as follow.
Comment 1: it is worth mentioning embolic strokes, which in most cases result from the same risk factors as thrombotic strokes.
Response to comment 1: Thank you for your precious comment. We totally agree with your advice. Patients with cardiogenic embolism also have a high rate of hospital recurrence, which is worthy of attention. As suggested, we have supplemented our Discussion section of the revised manuscript with the following:
“Another point worthy of attention is that patients with cardioembolic stroke met a high risk of in-hospital recurrence in both periods. The risk of early recurrence in the first two weeks after cardioembolic ischaemic stroke is between about 0.5% and 1.3% per day [27]. It seems that anticoagulant therapy is more widely used in our centre, but there was no statistically significant decrease in recurrence. Previous study revealed that ischemic stroke patients with atrial fibrillation shared similar risk factors with thrombotic stroke patients [28]. Meanwhile, risk factors for atherosclerosis including diabetes and hypertension are responsible for development of atrial fibrillation. Infec-tion is also related with atrial fibrillation in stroke patients [29]. Therefore, potentially undiagnosed atrial fibrillation may also contribute to the in-hospital recurrence. This reminds us that long-term electrocardiographic (ECG) monitoring needs to be more widely adopted.
[27] Seiffge, D.J.; Werring, D.J.; Paciaroni, M.; Dawson, J.; Warach, S.; Milling, T.J.; Engelter, S.T.; Fischer, U.; Norrving, B. Timing of anticoagulation after recent ischaemic stroke in patients with atrial fibrillation. Lancet Neurol 2019, 18, 117-126, doi:10.1016/S1474-4422(18)30356-9.
[28] Wankowicz, P.; Nowacki, P.; Golab-Janowska, M. Atrial fibrillation risk factors in patients with ischemic stroke. Arch Med Sci 2021, 17, 19-24, doi:10.5114/aoms.2019.84212.
[29] Suda, S.; Aoki, J.; Shimoyama, T.; Suzuki, K.; Sakamoto, Y.; Katano, T.; Okubo, S.; Nito, C.; Nishiyama, Y.; Mishina, M.; et al. Stroke-associated infection independently predicts 3-month poor functional outcome and mortality. J Neurol 2018, 265, 370-375, doi:10.1007/s00415-017-8714-6.”
Comment 2: it is worth referring to the manuscript which emphasizes the importance of statins in relation to patients with ischemic stroke and atrial fibrillation.
Response to comment 2: Thank you for your precious suggestion. The use of statins is an important part in the secondary prevention of ischaemic stroke. Observational studies have shown that in patients with symptomatic carotid artery stenosis, the use of statins is associated with a reduced risk of early ischemic stroke in patients with transient ischemic attack[1]. In‐hospital statin initiation was linked to better early stroke outcomes[2]. However, there is no high-quality research to analyze the safety and effectiveness of high-dose statins in reducing the risk of early recurrence of ischemic stroke[3]. But on the other hand, the improvement in secondary prevention is an important way to reduce the recurrence of stroke and improve the prognosis of stroke patients. Taken together, we added the discussion concerning the improvement of secondary prevention in our centre, including the use of statins. The corresponding added contents are as followings:
“Our research also shows progress in comprehensive stroke management. The patient received more aggressive treatment and risk factor screening examinations. Although statins do not appear to be effective at preventing all types of strokes or reducing all-cause mortality, they might reduce the risk of recurrent ischemic events with previous attacks [17]. In‐hospital statin initiation was linked to better early stroke outcomes [18]. Early initiation of existing treatments after minor stroke could reached an 80% reduction in the risk of early recurrence, which included antiplatelet therapy, the use of statin and blood pressure control [19]. Therefore, the clinical practice in our center reflected effects of improved secondary prevention on the decrease of in-hospital recurrence.”
Reference in this reply:
- Merwick, A.; Albers, G.W.; Arsava, E.M.; Ay, H.; Calvet, D.; Coutts, S.B.; Cucchiara, B.L.; Demchuk, A.M.; Giles, M.F.; Mas, J.L.; et al. Reduction in early stroke risk in carotid stenosis with transient ischemic attack associated with statin treatment. Stroke 2013, 44, 2814-2820, doi:10.1161/STROKEAHA.113.001576.
- Hong, K.S.; Lee, J.S. Statins in Acute Ischemic Stroke: A Systematic Review. J Stroke 2015, 17, 282-301, doi:10.5853/jos.2015.17.3.282.
- Hankey, G.J. Secondary stroke prevention. Lancet Neurol 2014, 13, 178-194, doi:10.1016/S1474-4422(13)70255-2.
Reviewer 3 Report
The authors report the incidence and associated factors of recurrent stroke in a retrospective single centre cohort in two different periods. Overall there is a large number of statistical analysis and details that distract the primary focus of the difference of incidence before and after an update of the local guidelines based on the CHANCE study.
There are several important limitations
- Why was this definition of recurrent stroke used? This is an definition and especially on the precedes the use of imaging to define TIA vs stroke. More modern definition specifically requires early recurrent stroke to be differentiated by the incident stroke by different territory or mechanism. Were these considered?
- Was imaging used to confirm the recurrent stroke (as opposed to extension or complications of the incident infarct)? This may be in the form of non-contrast or perfusion CT showing different territory, or MRI/DWI showing varying chronicity of the stroke.
- Was scoring recurrent stroke done prospectively during the patient's admission, or retrospectively (ie. going through images and clinical notes that are several years old)? If done retrospectively, how many neurologist was involved in scoring recurrent stroke and were there interrater variation estimation?
- Was multiple analysis used to correct for the large amount of statistical analysis?
- What is the outcome measure of table 5? Was there actually a change in practice between epoch 1 and epoch 2 among all patients? If there was actually no clinical change then it is invalid to conclude the drop in recurrent stroke is attributed to change in antiplatelets protocol.
- Baseline imaging and treatment are an important factors of stroke outcome. How many patients had thrombolysis? In particular, CT perfusion abnormalities in minor stroke portend poor prognosis further modulates treatment outcome. These points should be discussed.
- The authors should also discuss whether there are any other change in clinical practice that were relevant during the two periods. Eg. the adoption of thrombectomy after 2015-2016. Were there any difference in the rate of Carotid revascularisation in patients presenting with stroke and a symptomatic LAA?
Author Response
Dear reviewer, thank you for your careful review and constructive suggestions regarding our manuscript. We have revised the manuscript (Manuscript ID: brainsci-1515981) in accordance with the comments and marked all the amends on our revised manuscript. The point-by-point responses are provided below.
Comment 1: Why was this definition of recurrent stroke used? This is a definition and especially on the precedes the use of imaging to define TIA vs stroke. More modern definition specifically requires early recurrent stroke to be differentiated by the incident stroke by different territory or mechanism. Were these considered?
Response to comment 1: Thank you for your constructive suggestion. As mentioned in the reference 13[1], this definition is most clinically valid, which can avoid underestimation of risk. Our original intention in using this definition was to avoid missing any potential patients. All patients who met a possible recurrence event had an imaging examination to clarify the cause of new symptom (recurrent ischaemic stroke, haemorrhagic transformation, edema and progressive stroke because of other causes). New neurological deficit caused by ischemic lesions in different vascular territory was considered as a recurrent ischaemic event. If the new ischaemic event was located at the same vascular region as the first event, two trained neurologists in our team were responsible for checking the relationship between the new clinical symptom and the new ischaemic lesion. New ischemic events in the same vascular territory were considered as recurrent stroke when they were clearly associated with new neurological symptoms and was not related to other caused mentioned above. We are very sorry for inaccurate expression in the Materials and Methods section and have added relevant statements in the section of Outcome measures. Our research is a retrospective study and all our data were collected from medical records in Electronic Medical Record System. The mechanism of recurrent stroke was not recorded separately, so it was not used as a basis. As this is a limitation of the study, we have included it in the limitations section. The specific modifications are as follows:
“The primary endpoint was recurrent ischaemic stroke during hospitalization. The definition of recurrent ischaemic stroke referred to previous research [13]: (1) a new acute neurological event with symptoms lasting >24 hours after the index stroke; (2) the new event occurring >24 hours after the onset of the index stroke; and (3) the event was not associated with other neurological or systemic causes, including but not limited to haemorrhagic transformation, oedemaedema, seizure, heart failure, severe infection and so on. All patients who met a new acute neurological event had an imaging examination (non-contrast CT or MRI) to clarify the causes. New neurological deficit caused by ischemic lesions in different vascular territory was considered as a recurrent ischaemic event. It was also considered as a recurrent stroke if the new ischaemic event was located at the same vascular region as the index stroke and was associated with the new symptom clearly. Outcome measure was performed by two independent neurologists.”
“Third, we can only use the information in electronic medical records, as a result, some important information such as the mechanism of recurrent stroke was not included.”
Comment 2: Was imaging used to confirm the recurrent stroke (as opposed to extension or complications of the incident infarct)? This may be in the form of non-contrast or perfusion CT showing different territory, or MRI/DWI showing varying chronicity of the stroke.
Response to comment 2: Thank you for your comment! As mentioned above, all patients who met a new acute neurological event had an imaging examination. We used non-contrast CT or MRI/DWI to clarify to recurrent events. We have supplemented the description in the Materials and Methods section. Perfusion CT was not a common assessment at that time in our centre and was not included in the data collection and analysis. Since the perfusion CT plays a unique role in the stroke diagnosis and the assessment of stroke recurrence, we have made relevant supplements in the Discussion section. The corresponding added contents are as followings:
“Fourth, the lack of detailed imaging data and vascular assessment results is another important limitation. Infarcts in different periods, multiple acute infarcts, lesions with different circulations, and isolated cortical lesions all indicate a higher risk of early recurrence [8]. Minor stroke patients with acute large vessel occlusion are at risk of early recurrence which may be caused by hypoperfusion [33]. And CTP changes can be used to predict subsequent ischaemic tissue injury on DWI in TIA/minor stroke patients and have been shown to be of value in predicting recurrence [34]. Therefore, the imaging features for index stroke and evaluation vascular stenosis should strongly be considered in subsequent studies.”
Comment 3: Was scoring recurrent stroke done prospectively during the patient's admission, or retrospectively (ie. going through images and clinical notes that are several years old)? If done retrospectively, how many neurologists was involved in scoring recurrent stroke and were there interrater variation estimation?
Response to comment 3: Thanks for kind reminding of this. Our study a retrospective based on the data from medical records in Electronic Medical Record Management System, and the records was written by trained neurologists in our centre. Two trained neurologists reviewed the medical records (related narration was added in the revised manuscript). If the key word (“recurrence” in Chinese) or a new stroke event with an exact neurological deficit was recorded, we then analysed the results of imaging examination. The supervisor of our study was responsible to confirm the diagnosis for uncertain cases, which is not a common situation. We did not analyse the variation between two estimators during the study, which is a flaw in our research process. But the team members have no objections to the final diagnosis.
Comment 4: Was multiple analysis used to correct for the large amount of statistical analysis?
Response to comment 4: Thank you for your professional guidance. Statistical analysis of this study has certain limitations. Bonferroni correction for multiple testing was performed where applicable. Asterisks (*) indicate correlations surviving Bonferroni multiple testing correction and pounds (#) represents a p > 0.05 after correction for multiple comparison. Revision was made in the text and supplementary file. For the results of the subgroup analysis, we adopted a more conservative interpretation.
Comment 5: What is the outcome measure of table 5? Was there actually a change in practice between epoch 1 and epoch 2 among all patients? If there was actually no clinical change then it is invalid to conclude the drop in recurrent stroke is attributed to change in antiplatelets protocol.
Response to comment 5: Thank you for your kind reminding and question. The outcome measure of table 5 is in-hospital recurrence and we have revised the table title. Secondary prevention and auxiliary examination for cause screening have been improved over time. We did supplementary analysis and the results was contained in the Supplementary Table S1. This is also explained in the main text. We indeed found that the changes in treatment between two periods existed. Although there was no significant correlation between treatment and recurrence in univariate and multivariate analysis, the purpose of this study is to assess the clinical characteristics, risk factors and main treatments for in-hospital ischaemic recurrence in AIS patients over time. As a result, we mainly described the decline in recurrence rate in specific subgroups, pointed out the trend of treatment changes and related influencing factors.
Table S1. The comparison of treatment and auxiliary examination between different stages
|
|
Stage 1 (n=978) |
Stage 2 (n=1047) |
P value |
|
Treatment |
|
|
|
|
IVT (%) |
87 (8.9) |
52 (5.0) |
0.001* |
|
EVT in 24h (%) |
7 (0.7) |
12 (1.1) |
0.315 |
|
AntiPlt (%) |
939 (96.0) |
1001 (95.6) |
<0.001 |
|
None (%) |
39 (4.0) |
46 (4.4) |
0.649 |
|
Mono (%) |
551 (56.3) |
290 (27.7) |
<0.001* |
|
Dual (%) |
388 (39.7) |
711 (67.9) |
<0.001* |
|
Anticoagulant (%) |
35 (3.6) |
92 (8.8) |
<0.001* |
|
AntiHTN (%) |
474 (48.5) |
666 (63.6) |
<0.001* |
|
AntiDM (%) |
276 (28.2) |
373 (35.6) |
<0.001* |
|
Statin (%) |
890 (91.0) |
1028 (98.2) |
<0.001* |
|
Examination |
|
|
|
|
Holter (%) |
79 (8.1) |
543 (51.9) |
<0.001* |
|
UCG (%) |
909 (92.9) |
995 (95.0) |
0.048* |
|
CT (%) |
815 (83.3) |
597 (57.0) |
<0.001* |
|
MRI (%) |
940 (96.1) |
981 (93.7) |
0.014* |
|
CTA/MRA/DSA (%) |
868 (88.8) |
998 (95.3) |
<0.001* |
|
CVUS (%) |
895 (91.5) |
976 (93.2) |
0.148 |
* - significant difference (Bonferroni-adjusted P<0.0167 for AntiPlt). IVT, intravenous thrombolysis; EVT, endovascular treatment; AntiPlt, antiplatelet; AntiHTN, anti-hypertension agents; AntiDM, anti-diabetes agents; UCG, ultrasonic cardiogram; CTA, computed tomography angiography; MRA, magnetic resonance angiography; DSA, digital subtraction angiography; CVUS, carotid vessel ultrasound.
Comment 6: Baseline imaging and treatment are an important factors of stroke outcome. How many patients had thrombolysis? In particular, CT perfusion abnormalities in minor stroke portend poor prognosis further modulates treatment outcome. These points should be discussed.
Response to comment 6: We would like thank you for your important comments. Baseline imaging features are important predictors of stroke recurrence. Unfortunately, we lacked complete patient imaging data in this study. CT perfusion is not a routine examination in our center, therefore related data is not available due to the retrospective study design. In the current study, we mainly focused on the relationship between clinical features and the outcome, imaging characteristics could be explored in future studies. These above contents have been added as one of the limitations of our study. The additional text now reads as follows:
“Fourth, the lack of detailed imaging data and vascular assessment results is another limitation. Infarcts in different periods, multiple acute infarcts, lesions with different circulations, and isolated cortical lesions all indicate a higher risk of early recurrence [8]. Minor stroke patients with acute large vessel occlusion are at risk of early recurrence which may be caused by hypoperfusion [33]. And CTP changes can be used to predict subsequent ischaemic tissue injury on DWI in TIA/minor stroke patients and have been shown to be of value in predicting recurrence [34]. Therefore, the imaging features for index stroke and evaluation vascular stenosis should strongly be considered in subsequent studies.”
Comment 7: The authors should also discuss whether there are any other change in clinical practice that were relevant during the two periods. Eg. the adoption of thrombectomy after 2015-2016. Were there any difference in the rate of Carotid revascularisation in patients presenting with stroke and a symptomatic LAA?
Response to comment 7: Thank you for your advice. A detailed comparison to the treatments between different period is presented in Supplementary Table 1. Reperfusion therapy, including IVT and EVT, is imperative to improve stroke outcome. However, the number of patients with reperfusion therapy is limited. The improvement of reperfusion therapy is not obvious in our centre as well. The probability of recurrence event after IVT was relatively small based on previous study[2,3]. At the same time, very early recurrence after reperfusion therapy would be excluded based on the definition of recurrent stroke we used. As a result, insufficient number of patients caused difficulties in further analysis. Those are indeed the shortcomings of our study and have been highlighted in the discussion. Further analysis needs to be broadened to a larger sample size and a more detailed therapy regimen. The study limitations were described as follows:
“This study has its limitations. First, our study was a retrospective, single-centre design and lacked out-of-hospital follow-up information. Our findings need to be further confirmed in a study with a larger sample size and prospective research. Second, the assessment of antiplatelet therapy was a crude approach. The results of vascular evaluation are also important in choosing an antiplatelet regimen, and these were not included during the grouping process. Third, we can only use the information in electronic medical records, as a result, some important information such as the mechanism of recurrent stroke was not included. The evaluations of other risk factors, treatments, comorbidities and complications are also insufficient. In Stage 2, the prescription of anticoagulants seemed to be related to ischaemic recurrence. However, accurate records of anticoagulation treatment after antiplatelet therapy are not available to us for all cardioembolic stroke cases; thus, we did not analyse the relationship between treatments and in-hospital recurrence in cardioembolic patients. Compared with Stage1, more patients in Stage 2 received antihypertensive or hypoglycemic treatment. Further analysis needs to be broadened to include a more detailed therapy regimen and their effect on the outcome event. Fourth, the lack of detailed imaging data and vascular assessment results is another limitation. Infarcts in different periods, multiple acute infarcts, lesions with different circulations, and isolated cortical lesions all indicate a higher risk of early recurrence [8]. Minor stroke patients with acute large vessel occlusion are at risk of early recurrence which may be caused by hypoperfusion [33]. And CTP changes can be used to predict subsequent ischaemic tissue injury on DWI in TIA/minor stroke patients and have been shown to be of value in predicting recurrence [34]. Therefore, the imaging features for index stroke and evaluation vascular stenosis should strongly be considered in subsequent studies”
Reference in this reply:
- Coull, A.J.; Rothwell, P.M. Underestimation of the early risk of recurrent stroke: evidence of the need for a standard definition. Stroke 2004, 35, 1925-1929, doi:10.1161/01.STR.0000133129.58126.67.
- Kamal, H.; Mowla, A.; Farooq, S.; Shirani, P. Recurrent ischemic stroke can happen in stroke patients very early after intravenous thrombolysis. Journal of the Neurological Sciences 2015, 358, 496-497, doi:10.1016/j.jns.2015.09.020.
- Awadh, M.; MacDougall, N.; Santosh, C.; Teasdale, E.; Baird, T.; Muir, K.W. Early Recurrent Ischemic Stroke Complicating Intravenous Thrombolysis for Stroke Incidence and Association With Atrial Fibrillation. Stroke 2010, 41, 1990-1995, doi:10.1161/Strokeaha.109.569459.
Round 2
Reviewer 3 Report
The revision has improved the manuscript.
- The methodology needs to be specified that recurrent stroke incidence was made by retrospective chart review, and that imaging was used in when the new symptoms occurred in the same territory.
- Please add justification why Bonferroni corrections only divided the p value by 2.
- The authors have included thrombolysis/treatment rates in a table but has not specifically discussed this in the manuscript. This is particularly relevant in the topic of minor stroke as clinical decision making for thrombolysis/thrombectomy in minor stroke is controversial and hence can vary significantly. CTP can be used to assess extent and severity of hypoperfusion to guide thrombolysis. This should probably be discussed and included as an additional limitation (ie that CTP was not available to guide IVT treatment and stratify risk of poor outcome).
- Please make it clear for the readers how the different time epochs were defined (ie. update of the Chinese guidelines). Also include brief discussion about whether there were any important changes in treatment guidelines for secondary prophylaxis (apart from antiplatelet uses) that may be explained some of the differences observed in the two epochs.
- There remains some hypos. Please correct.
Author Response
Dear Reviewer: Thank you again for taking the time to review this paper a second time. We have modified the manuscript in accordance with your valuable and constructive comments and marked all the amends on our revised manuscript. My explanation to the comments point- by- point is as follow:
Comment 1: The methodology needs to be specified that recurrent stroke incidence was made by retrospective chart review, and that imaging was used in when the new symptoms occurred in the same territory.
Response to comment 1: Thank you for your feedback. The description of the method is still not sufficiently detailed. The manuscript has been changed accordingly, and the way to determine the main outcome has been described in more detail, including its retrospective design and the imaging evaluation. The sentence has been changed as indicated below:
“The primary endpoint was recurrent ischaemic stroke during hospitalization. Recurrent ischaemic stroke was defined as follows according to previous research: (1) a new acute neurological event with symptoms lasting >24 hours after the index stroke; (2) a new event occurring >24 hours after the onset of the index stroke; and (3) an event not associated with other neurological or systemic causes, including but not limited to haemorrhagic transformation, oedema, seizure, heart failure, severe infection, etc. Clinical information for each patient was obtained from retrospective review using our electronic medical record system. All patients with a new acute neurological event underwent an imaging examination (noncontrast CT or MRI) to identify the cause. New neurological deficits caused by ischaemic lesions in different vascular territories were considered recurrent ischemic events. An event was also considered a recurrent stroke if the new ischaemic lesion on the imaging was located in the same vascular region as the index stroke and was clearly associated with a new symptom. The outcome measurements were performed by two independent neurologists.”
Comment 2: Please add justification why Bonferroni corrections only divided the p value by 2.
Response to comment 2: Thank you for your comment. Bonferroni correction method was used to perform multiple testing correction. We divided the antiplatelet treatment regimens into 3 groups (negative, neutral, positive) and analysed related influencing factors. We used the neutral group as a reference. As a result, we made comparisons between negative and neutral groups as well as between positive and neutral groups. Statistical comparisons were made twice, so we divided the p value by 2 in this section.
Comment 3: The authors have included thrombolysis/treatment rates in a table but has not specifically discussed this in the manuscript. This is particularly relevant in the topic of minor stroke as clinical decision making for thrombolysis/thrombectomy in minor stroke is controversial and hence can vary significantly. CTP can be used to assess extent and severity of hypoperfusion to guide thrombolysis. This should probably be discussed and included as an additional limitation (ie that CTP was not available to guide IVT treatment and stratify risk of poor outcome).
Response to comment 3: Thank you so much for your great suggestion. No correlation was found between early reperfusion therapy and in-hospital recurrence in this study and in some previous studies[1,2], so we did not discuss this issue in detail. On the other hand, however, early reperfusion is of unequivocal benefit. This point therefore deserves to be discussed further. There is controversy regarding whether minor stroke patients can benefit from IVT. Advanced imaging methods, including CTP, can be used to guide thrombolytic therapy[3]. However, CTP was not carried out as a routine examination in our centre during the study, which may influence the judgement for the risk of in-hospital recurrent stroke and the treatment choices. An addition has been made to the discussion and limitation. The details of revisions are as follows:
“However, the number of patients with reperfusion therapy is limited. Improvement in reperfusion therapy is also not obvious in our centre. Although the current study did not show any association between early reperfusion therapy and in-hospital recurrence in atherosclerotic stroke or minor stroke patients, reperfusion is still the primary goal of stroke treatment. Therefore, promotion of reperfusion therapy should be prioritized for improvement in future development efforts.”
“Fourth, the lack of detailed imaging data and vascular assessment results is another important limitation. Infarcts in different periods, multiple acute infarcts, lesions with different circulations, and isolated cortical lesions all indicate a higher risk of early recurrence. Minor stroke patients with acute large vessel occlusion are at risk of early recurrence, which may be caused by hypoperfusion. CTP changes can be used to predict subsequent ischaemic tissue injury on DWI in TIA/minor stroke patients and have been shown to be of value in predicting recurrence. The lack of relevant evaluation may lead to under-identification of risk populations. Those patients may have been undertreated in our study. The lack of advanced imaging may also deny a segment of patients access to a potentially beneficial reperfusion treatment. Therefore, the imaging features of the index stroke and evaluation of vascular stenosis should strongly be considered in subsequent studies.”
Comment 4: Please make it clear for the readers how the different time epochs were defined (ie. update of the Chinese guidelines). Also include brief discussion about whether there were any important changes in treatment guidelines for secondary prophylaxis (apart from antiplatelet uses) that may be explained some of the differences observed in the two epochs.
Response to comment 4: Thank you so much for your great suggestion. Early dual antiplatelet therapy for minor stroke patients is one of the most important updates in the edition of the Guideline. Therefore, we mainly discussed the change in antiplatelet treatment regimen during the study during. Another curial update is to standardize the extended time window, indications and contraindications for intravenous thrombolysis. However, we could not find significant relationship between IVT and in-hospital recurrence. Thus, we did not make a detailed discussion initially. Besides, the updated guideline also provides guidance on blood pressure target, blood sugar management and neuroprotective therapy. In view of your opinion, we have revised the Materials and Methods and Discussion sections. The description of grouping criteria has been rephrased. The improvement of other treatments has been discussed in a separate paragraph. The specific modifications are as follows:
“Our study was a retrospective, single-centre analysis based on patients admitted to the Neurology Department of Peking University Third Hospital between January 2014 and December 2017. The new version of the “Guidelines for the Diagnosis and Treatment of Acute Ischaemic Stroke in China” was published in 2015, as a result, the data were divided into 2 groups according to this timepoint. The patients in Stage 1 were admitted from January 2014 to December 2015, while the Stage 2 patients were admitted from January 2016 to December 2017.”
“Our research also shows progress in comprehensive stroke management. The new version of Chinese stroke guideline pointed out the importance of treatment in acute phase. The guideline standardized the extended time window, indications and contraindications for intravenous thrombolysis. Apart from the early use of dual antiplatelet therapy, other treatments including anticoagulant therapy, blood pressure management, blood glucose management and the use of neuroprotective agents were also updated. In our centre, the patients received more aggressive treatments and risk factor screening examinations in the later period. Although statins do not appear to be effective in preventing all types of strokes or reducing all-cause mortality, they might reduce the risk of recurrent ischaemic events with previous attacks. In‐hospital statin initiation was linked to better early stroke outcomes The early initiation of existing treatments, including antiplatelet therapy, statins and blood pressure control, after minor stroke could result in an 80% reduction in the risk of early recurrence. Some drug trials have revealed a beneficial effect of intensive glycaemic control on risk of stroke although further high-quality studies are required to confirm that. Therefore, the clinical practice in our centre reflected the effects of improved secondary prevention on the decrease in in-hospital recurrence……”
Comment 5: There remains some hypos. Please correct.
Response to comment 5: Thank you for your suggestion. The revised manuscript has been polished by a professional language academic polishing company.
Reference:
- Yu, F.; Liu, X.; Yang, Q.; Fu, Y.; Fan, D. In-hospital recurrence in a Chinese large cohort with acute ischemic stroke. Sci Rep 2019, 9, 14945, doi:10.1038/s41598-019-51277-8.
- Erdur, H.; Scheitz, J.F.; Ebinger, M.; Rocco, A.; Grittner, U.; Meisel, A.; Rothwell, P.M.; Endres, M.; Nolte, C.H. In-hospital stroke recurrence and stroke after transient ischemic attack: frequency and risk factors. Stroke 2015, 46, 1031-1037, doi:10.1161/STROKEAHA.114.006886.
- Thomalla, G.; Boutitie, F.; Ma, H.; Koga, M.; Ringleb, P.; Schwamm, L.H.; Wu, O.; Bendszus, M.; Bladin, C.F.; Campbell, B.C.V.; et al. Intravenous alteplase for stroke with unknown time of onset guided by advanced imaging: systematic review and meta-analysis of individual patient data. Lancet 2020, 396, 1574-1584, doi:10.1016/S0140-6736(20)32163-2.
- Rothwell, P.M.; Giles, M.F.; Chandratheva, A.; Marquardt, L.; Geraghty, O.; Redgrave, J.N.; Lovelock, C.E.; Binney, L.E.; Bull, L.M.; Cuthbertson, F.C.; et al. Effect of urgent treatment of transient ischaemic attack and minor stroke on early recurrent stroke (EXPRESS study): a prospective population-based sequential comparison. Lancet 2007, 370, 1432-1442, doi:10.1016/S0140-6736(07)61448-2.